# Effect of the Type of Inorganic Binder on the Microstructure and Properties of AlSi7Mg Alloy Castings Made by Ablation Casting Technology

**DOI:** 10.3390/ma15144912

**Published:** 2022-07-14

**Authors:** Jadwiga Kamińska, Michał Angrecki, Sabina Puzio

**Affiliations:** Łukasiewicz Research Network-Krakow Institute of Technology, Zakopiańska 73, 30-418 Krakow, Poland; michal.angrecki@kit.lukasiewicz.gov.pl (M.A.); sabina.puzio@kit.lukasiewicz.gov.pl (S.P.)

**Keywords:** casting technology, aluminium alloys, inorganic binders, microstructure examinations, strength tests

## Abstract

The results of studies on the effect that the type of binder and casting technology exert on the microstructure and properties of AlSi7Mg alloy castings are discussed in this paper. Comparative tests were carried out on three casting manufacturing technologies, i.e., conventional sand mould casting and cooling process, metal mould (die) casting, and sand mould casting with ablation breakdown of mould and cooling of castings. Moulds were made from four different sand mixtures with inorganic binders hardened by various technologies. The microstructure of test castings was examined at three different levels, i.e., in the upper part, central part, and lower part of each casting. The tensile test at room temperature was carried out in accordance with standards. The experimental results showed differences in the microstructure of castings. The differences resulted from changes in the crystallisation path due to the use of three different casting technologies, ensuring different rates of heat dissipation from castings; they were also due to the shape of castings. It has been shown that castings made by ablation technology are characterised by a high degree of the microstructure refinement (SDAS reduced by 18–30%), which gives higher strength properties than the properties of castings made in conventional sand moulds. Samples taken from castings made by the ablation technology in moulds with phosphate binder and microwave-hardened geopolymer binder were characterised by the mechanical properties comparable to gravity die castings.

## 1. Introduction

The main indicator of economic development in the foundry industry is the share of castings made from non-ferrous metals, mainly light metal alloys, in the total production of castings [1]. An upward trend in the production of light alloy castings is forecast for 2020–2025 [2,3]. This is due to the growing demand for lightweight structures, especially in the transport industry—automotive and aviation—where the need for reduced fuel consumption is dictated not only by economic factors but also by ecological considerations. There is a tendency in the market to replace elements made of cast steel and cast iron with light alloys which, owing to their high casting and mechanical properties, can be used in a wider range of applications. The most common methods of casting aluminium alloys are pressure die casting, gravity die casting, and gravity casting into sand moulds, including ablation casting technology [1,4,5]. Ablation casting technology is a sand casting process based on inorganic binders, where the sand moulds are intensively cooled by an ablation agent (usually water) [6,7]. This results in a high temperature gradient and a short solidification time, both of which are highly desirable, if a fine-grained structure and properties comparable to castings made in metal moulds are to be obtained. The technology is primarily intended for castings with variable wall thickness and complex shapes, moulded without the use of foundry cores.

The aim of this study was to show the effect of the type of binder, the method of its hardening, and the type of casting technology on the microstructure and properties of castings made from the AlSi7Mg (AK7) alloy. Comparative tests were carried out on three casting technologies, i.e., sand casting with traditional cooling, casting into metal moulds (dies), and sand casting with ablation-assisted mould breaking and casting cooling.

The ablation casting technology allows for the production of high-quality castings free from porosity and impurities and with the required high strength properties. An additional advantage of the ablation process is the possibility of recovering the base sand grains without the need to carry out the process of mechanical reclamation. During the process of removal of casting from the mould, the binder is washed off from the sand grains. The binder is dissolved in water, while the sand grains after drying have the properties similar to fresh silica sand. In line with the trends observed in recent years, moulding processes must meet stringent requirements related to environmental protection, including problems with the disposal of waste moulding materials [8]. There is one technological solution in the world, developed by Alotech Ltd., LLC (Brooklyn, OH, USA) and patented in 2008 [9]. This technology has been put into practical use by the American Company Mercury Marine (Fond du Lac, WI, USA) and by Acura (Honda) (Hamamatsu, Japan) [10].

The moulding sand for the manufacture of moulds used in the ablation casting process should have the strength sufficient to withstand the metallostatic pressure of molten alloys, while being at the same time easy to disintegrate with the ablation agent. Earlier studies by the authors showed that the bending strength of the moulding sands intended for ablation casting should be in the range of 1.5–1.7 MPa [11,12,13,14]. The latest trends in foundry technology focus on the improvement in dimensional accuracy and functional quality of castings with regard to the environmental protection requirements introduced in the European Union. Successive generations of moulding materials are being developed, including inorganic binders, along with the technologies for their hardening, to allow for a reduced binder content in the sand, while maintaining the best strength properties [15,16]. 

Casting quality and properties are affected by the phenomena that occur at the metal-mould interface. The solidification rate of metal in foundry moulds depends on the following factors: type of alloy, degree of overheating, and thermophysical properties of mould determined by the heat accumulation coefficient [17]. An improvement in the mechanical properties of Al-Si alloys is obtained mainly by changing the morphology of precipitates through alloy modification, maintaining the required overheating and pouring temperature, upgrading the heat treatment parameters and increasing the heat dissipation rate from castings. One of the possibilities is the erosive breakdown of the mould immediately after it has been poured with molten metal. The direct contact between molten metal and mould material without the formation of a gas gap accelerates the transfer of heat from the metal, significantly increasing the rate of casting cooling and solidification [6,9]. In 2008, Alotech Ltd. LLC (Brooklyn, OH, USA) developed a new technology of casting into sand moulds intensively cooled with water during casting solidification and disintegrated after the end of the solidification process. The technology is referred to as ablation casting [9]. The principle of the process operation was described in detail by J. Grassi et al. [6] and P. Dudek et al. [18,19]. The process is applicable to aluminium and magnesium alloys. V. Bohlooli et al. [20] investigated the effect of process variables on the porosity and microstructure of A356 casting alloy. D. Sui and Q. Han [21] attempted to model the A356 alloy casting process by ablation. The team of A.I. Fernández-Calvo [22] conducted some trials with the ablation casting of A356 alloys. The study compares the results of microstructural examinations and mechanical testing of the T6 heat-treated castings made by three technologies, i.e., ablation casting, conventional sand mould casting, and metal mould (die) casting. M. Ghasvari and M.A. Boutorabi [22] conducted ablation casting trials of AZ81 magnesium alloys. 

The main objective of the research presented in this article was to examine the effect of cooling rate on the microstructure of casting at three different levels, i.e., in the upper part, central part, and lower part of casting, and interrelate the results obtained with the type of moulding material and casting technology. The dispersion of the dendritic structure and the secondary dendrite arm spacing (SDAS) were also determined. The tensile test at room temperature was carried out in accordance with the Polish Standard PN-EN ISO 6892-1:2016-09. 

## 2. Materials and Methods

Moulds were made from four different sand mixtures based on fresh Sibelco silica sand with the additions of inorganic binders hardened by different technological processes, i.e., thermally hardened water glass R150 and thermally hardened geopolymer binder, microwave-hardened geopolymer binder, and MgO-hardened phosphate binder used in the technology of self-hardening loose sand mixtures. Sibelco silica sand classified as medium by the Polish Standard PN-85/H-11001 [23] (main fraction 0.20/0.16/0.315) was used as a base material of the sand mixtures. Moulding materials were processed in an LM-R1^®^ Multiserw Morek (Marcyporęba, Poland) laboratory mixer, using the following mixing times: for self-hardening sand: sand and hardener—90 s and for sand, hardener, and binder—another 90 s; for thermally and microwave-hardened sands: sand and binder—90 s. From the sand mixtures, standard oblong samples with dimensions of 22.36 × 22.36 × 172 mm were made for the bending test. The sand was pre-compacted in a device for vibratory compaction of samples, Multiserw Morek (Marcyporęba, Poland) type LUZ-1^®^. The vibration time was 20 s, followed by hardening in air (Sand No. 1), in a microwave chamber with waves of 800 W power for 6 min (Sand No. 2), and thermally at 200 °C for 10 min (Sands No. 3 and No. 4). Bending strength measurements were carried out on an LRu-2e^®^ apparatus for the determination of strength properties made by Multiserw Morek (Marcyporęba, Poland). Each time, measurements were made on three samples and the result was an arithmetic mean of the results not deviating from the mean by more than 10%. Erosion tests were performed using a device for the removal of moulding sand from casting and the device for casting cooling [24].

The test castings were made in moulds prepared on the basis of the four previously mentioned moulding materials: on the basis of a phosphate binder cured with magnesium oxide (moulding sand No. 1), on the basis of geopolymer binder cured with microwaves (moulding sand No. 2), on the basis of sodium water glass R150 thermally cured (moulding sand No. 3), and a geopolymer binder thermally cured (moulding sand No. 4). The test castings were made from an AlSi7Mg aluminium–silicon alloy. The shape of the casting is shown in Figure 1. The pouring temperature was 720 °C. Additionally, the metal mould was designed to compare the effect of ablation casting technology on the course of casting solidification process. This allowed us to compare the three technologies of casting manufacture, i.e., traditional sand mould casting and cooling, metal mould (die) casting, and sand mould casting with ablation-assisted mould breakdown and casting cooling. 

Casting trials were carried out simultaneously, in a sand mould, in a die mould, and in a sand mould using the ablative casting technology, which is the sand mould intensively cooled with water. Thermocouples were attached to each mould, at three positions (in the upper, middle, and bottom part of the casting). This allowed for the assessment of the heat transfer from the castings.

The thermal analysis was carried out on two measuring devices. For traditional sand casting, the curves were recorded on the multi-channel MrAC-15^®^ (JOTA s.c., Cracow, Poland) recorder. The temperature curves of the ablative casting process were recorded on a portable TES-1384 (TES Electrical Electronic Corp., Taipei, Taiwan) recorder. This allowed us to record the solidification and cooling of the castings over time.

The ablative process start time was set at 1 min from the moment of the end of pouring. Shorter time was not sufficient to maintain the shape of the casting, while longer time did not guarantee significant changes in the structures of the castings.

Microstructural examinations were performed on three levels, i.e., in the upper part of the casting (No. 3 in Figure 1), in the central part (No. 2), and in the lower part (No. 1). The samples were polished on a Struers (Copenhagen, Denmark) grinding polisher according to the program for soft materials (using 220, 500, and 1000 papers with 9, 3, 1, and ¼ micron diamond pastes) and were next etched in a 1% HF solution in distilled water. The examinations were made using a Zeiss (Oberkochen, Germany) Observer.Z1m^®^ light microscope with the Axio Observer Zm10 program. SDAS was measured with a ZEISS AxioVision40 V 4.8.2.0 (ZEISS MicroImaging GmbH, Munich, Germany) program. Measurements were made on etched specimens at 50× magnification using the 10 oriented secant method. The static tensile test at ambient temperature was carried out in accordance with the Polish Standard PN-EN ISO 6892-1:2016-09 Method B [25]. The tests were carried out on a VEB Leipzig type EU-20^®^ (VEB Werkstoffprüfmaschinen Leipzig GmbH, Leipzig, Germany) testing machine with a maximum range of 200 kN. The stress rate increase was 15.9 MPa/s. Samples according to the Polish Standard PN-EN ISO 6892-1:2010 [26] were used in the tests.

Figure 2 shows examples of moulds poured with metal.

## 3. Results and Discussion

The moulds for the test castings were made on the basis of four moulding materials. The composition of the selected moulding sands and the values of the bending strength and mould erosion time are compared in Table 1. The selected moulding materials are characterised by good strength and short erosion time suitable for the ablative casting process. The parameters ensure quick contact of the coolant with the casting surface, and thus its quick cooling.

### 3.1. Thermal Analysis

The thermal analysis was used to show the differences in the casting crystallisation path resulting from the use of different casting technologies. The casting cooling curves represent the average of three measurements for all tested moulding materials and each casting technology. Results are shown in Figure 3.

In the case of the analysis of moulding material No. 1, the curve of heat transfer for the ablative casting is irregular. The casting was cooled very quickly, comparable to a die mould casting. The analysis of moulding materials No. 2 and 3 have a similar course. It can be noticed that although the initial heat transfer process from both sand castings is similar, from the moment of breaking the mould and loss of direct contact of the casting with the cooling agent, the casting rapidly reached the room temperature. In the case of the analysis of moulding material No. 4, the curve of heat transfer from the ablative casting is irregular, while cooling occurs very fast. After approximately 200 s, the casting reached the room temperature. The initial temperature drop was faster than the die mould casting.

The die mould in which the castings were made was not preheated; therefore, the castings quickly cooled to a temperature of approximately 300 °C.

### 3.2. Microstructure of Test Castings

The results of microstructural examinations made on samples taken from the central part of casting are presented. SDAS is shown for all three casting levels. Figure 4, Figure 5, Figure 6 and Figure 7 show the dispersion of casting structure at 50× magnification. The secondary dendrite arm spacing values are summarised in Table 2. 

Microscopic observations have shown that between the tested castings made by different technologies and between different zones in each casting (top, centre, bottom), there are differences in the dispersion of the dendritic structure, reflecting differences in the cooling rate. The differences are due to changes in the crystallisation path caused by the use of three different casting technologies producing different rates of heat dissipation from castings. They are also due to the casting shape, the top of which that is directly connected to the gating system is much wider than the bottom. The morphology of the dendritic structure, including the length of dendrites visible on the examined casting sections, is typical for a given casting technology. The shortest dendrites (approx. 400 µm) were produced by die casting, dendrites with a length of approx. 500–800 µm were formed in the ablation casting process, and the estimated length of the dendrites formed during sand mould casting was approx. 1000 µm. Detailed analysis of the microstructure images shows the high degree of both grain and eutectic refinement in castings made by the ablation process as compared to the traditional sand casting technology. This is also confirmed by the SDAS studies.

The examination and comparison of samples made by the ablation process and traditional sand casting process show a significant decrease in the SDAS values obtained for ablation castings (Table 2). Reducing SDAS affected the morphology of silicon precipitates (the eutectic phases were formed from fine-grained silicon). The finer the silicon phases, the more uniform the casting properties. Coarse-grained eutectic silicon phases can act as crack initiation zones and reduce the mechanical properties of aluminium castings. The maximum SDAS reduction was obtained for castings made in Sands No. 1 and No. 2. In the case of moulds made from Sand No. 1, the value of SDAS decreased by about 20% compared to the casting solidifying in a traditional way. The SDAS of the sand casting was over 35% higher than that of the die casting, but this difference dropped to 18% after the use of ablation cooling. A similar relationship was observed for moulds made from Sand No. 2. In the case of ablation castings poured into moulds made from Sands No. 3 and No. 4, the obtained properties were not so favourable, mainly because of the longer time of mould erosion. 

The morphology of eutectic silicon precipitates (Figure 8) and the phase constituents of microstructure (Figure 9) are shown for the casting made in Sand No. 4. Microscopic observations showed differences in the dispersion of eutectic Si precipitates, but no changes in the morphology of individual precipitates related to the casting technology or cooling rate were observed (Figure 8). According to the visual assessment, both sand casting and ablation casting had a comparable volume fraction of the interdendritic eutectic, while in the case of die castings, this fraction was clearly lower. The intermetallic phases in the interdendritic eutectic were determined from the local chemical composition (Figure 9). The occurrence of Mg_2_Si and Π-AlFeMgSi phases was observed in all samples. Locally, mainly in castings made in sand moulds, the β-AlFeSi and α-Al(Fe,Mn)Si phases also appeared.

### 3.3. Results of Standard Tensile Test

Table 3 shows the mechanical properties measured on samples taken from the test castings. The results are also presented graphically in Figure 10, Figure 11 and Figure 12. Based on the information contained in relevant standards [27], the obtained results indicate the material designated as EN-AC-42000 (EN AC-Al Si7Mg), which in the as-cast state should be characterised by the following properties: yield strength Rp0.2—min. 80 MPa, tensile strength Rm—min. 140 MPa, elongation A—min. 2%. 

Regarding samples taken from the castings poured in sand moulds, none of the proposed sand mixtures produced casting properties that would meet the standard requirements in terms of Rm and A. Rapid ablation cooling enabled obtaining castings with properties definitely superior to the properties of castings undergoing the traditional process of solidification in sand moulds. In the case of ablation castings made in moulds based on Sands No. 1 and No. 2, the obtained properties were similar to those of a die casting. For these sands, castings made by the ablation technology had over 45% higher tensile strength than that obtained in common sand castings. Both elongation A and strength were comparable to those of a die casting.

## 4. Conclusions

Analysing the results of the research described in this article, the following conclusions are drawn:The selection of the sand composition for ablation casting has a significant impact on both the microstructure and properties of the resulting castings. The particularly important parameter is the time of mould erosion, since it determines the rate of mould breakdown by the ablation agent, ensuring rapid heat dissipation from castings.Microstructural examinations showed differences in the microstructure of castings. The differences were due to changes in the crystallisation path resulting from the use of three different casting technologies, ensuring different rates of heat dissipation from castings, and from the use of four different moulding sand mixtures. A varied dispersion of eutectic Si precipitates is present, while no changes in the morphology of individual precipitates related to the cooling rate were observed.It has been shown that castings made by the ablation technology are characterised by the refined microstructure (SDAS reduced by 20–25%), which confers to them the strength properties superior to the properties obtained in castings made by the conventional sand moulding process.The best strength parameters in the ablation technology were obtained by castings made in the sand moulds based on a phosphate binder hardened with MgO (Sand No. 1) and on a microwave-hardened geopolymer binder (Sand No. 2). The erosion time of moulds made from these sand mixtures was 11 s and 12 s, respectively.For both sands, the obtained values of the tensile strength and elongation were comparable to those obtained for a die casting (for Sand No. 1: Rm −156 MPa, A—1.7% and for Sand No. 2: Rm = 156 MPa, A = 1.7%, for die castings: Rm—155 and 158 MPa, A—1.8 and 1.9%).

Taking into account the strength parameters of the castings and the environmental aspects, the most recommended technology for the ablation casting process is the technology using moulds based on a thermally hardened geopolymer binder (Sand No. 2). The conducted research has shown that the ablation casting technology is an effective and low-cost method to obtain the desired mechanical properties in the aluminium foundry industry.

## Figures and Tables

**Figure 1 materials-15-04912-f001:**
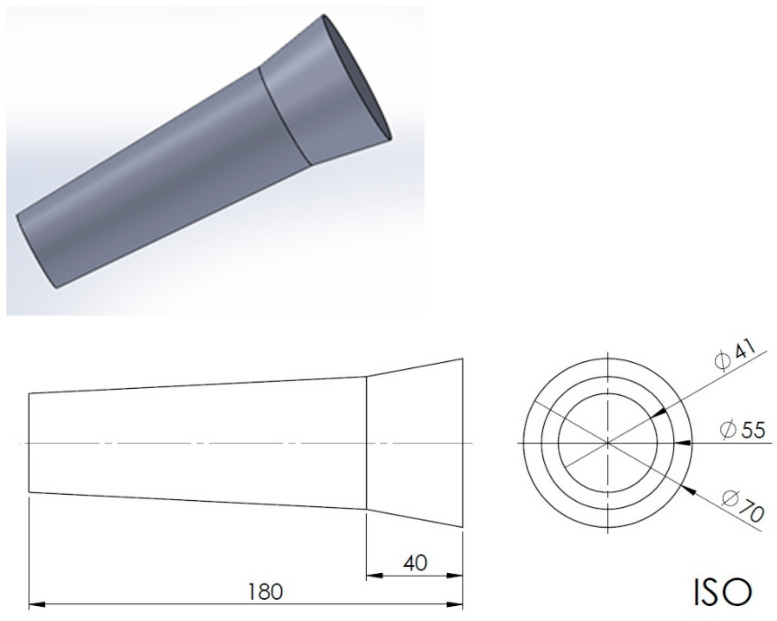
Schematic representation of the test casting.

**Figure 2 materials-15-04912-f002:**
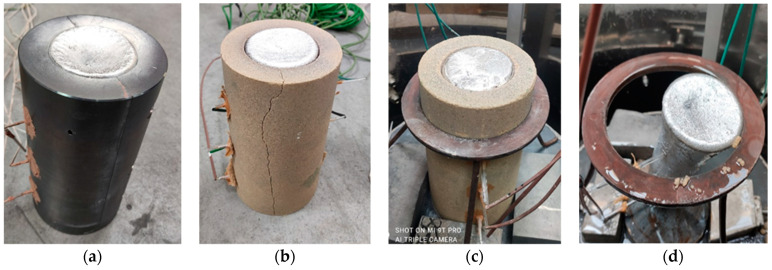
Moulds poured with the AlSi7Mg alloy: (**a**) metal mould, (**b**) standard sand mould poured with metal, (**c**) sand mould poured in ablation casting technology, (**d**) casting after mould removal by ablation.

**Figure 3 materials-15-04912-f003:**
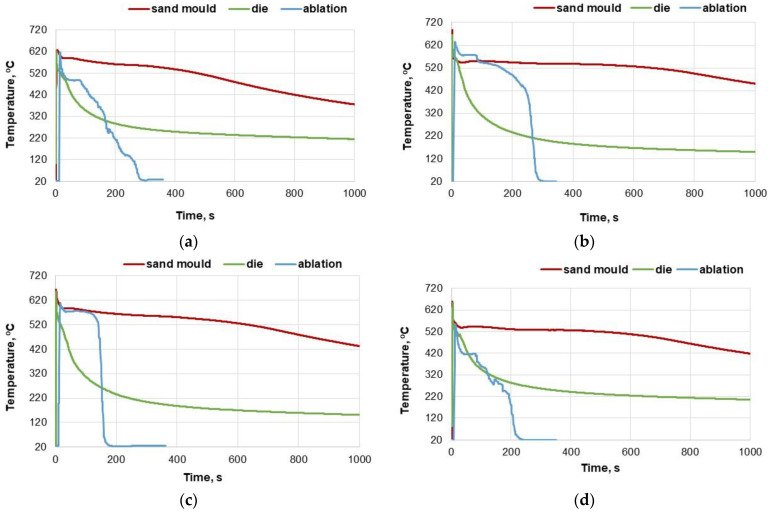
Comparison of cooling curves (average for three positions), for each of the presented technologies: (**a**) Sand No. 1, (**b**) Sand No. 2, (**c**) Sand No. 3, (**d**) Sand No. 4.

**Figure 4 materials-15-04912-f004:**
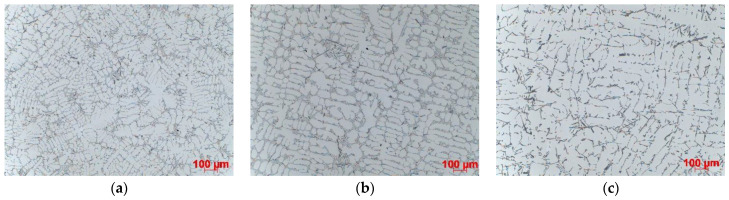
Dispersion of dendritic structure in casting made in Sand No. 1, etched with HF, 100× magnification: (**a**) die, (**b**) ablation, (**c**) sand mould.

**Figure 5 materials-15-04912-f005:**
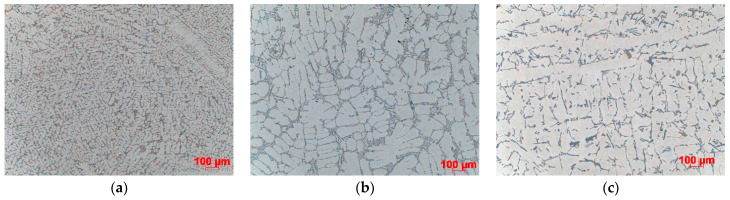
Dispersion of dendritic structure in casting made in Sand No. 2, etched with HF, 100× magnification: (**a**) die, (**b**) ablation, (**c**) sand mould.

**Figure 6 materials-15-04912-f006:**
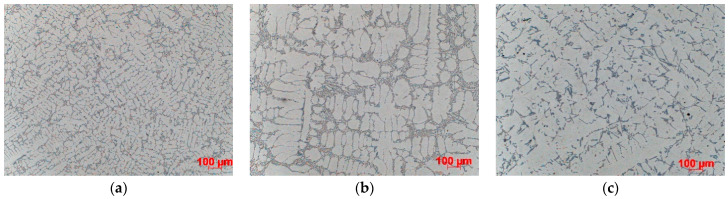
Dispersion of dendritic structure in casting made in Sand No. 3, etched with HF, 100× magnification: (**a**) die, (**b**) ablation, (**c**) sand mould.

**Figure 7 materials-15-04912-f007:**
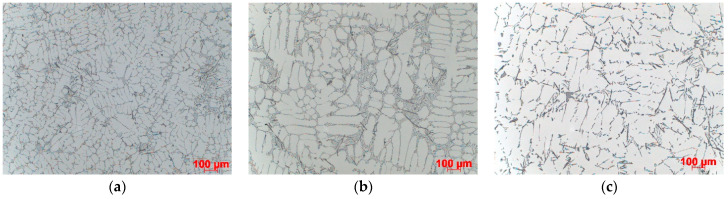
Dispersion of dendritic structure in casting made in Sand No. 4, etched with HF, 100× magnification: (**a**) die, (**b**) ablation, (**c**) sand mould.

**Figure 8 materials-15-04912-f008:**
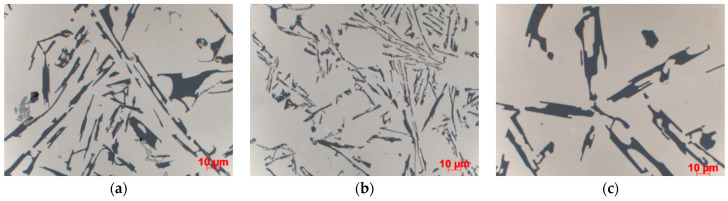
Morphology of eutectic silicon in casting made in Sand No. 4, centre, LM, etched with HF, 1000× magnification; (**a**) die, (**b**) ablation, (**c**) sand mould.

**Figure 9 materials-15-04912-f009:**
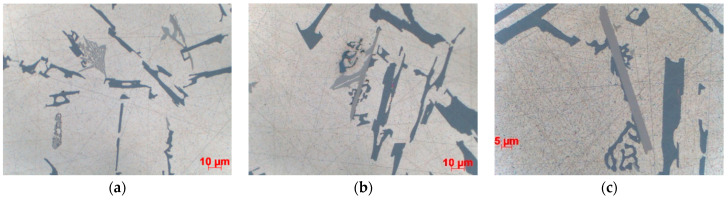
Phase constituents of the microstructure of casting made in Sand No. 4, LM; (**a**) ablation—top, (**b**) sand—top, (**c**) sand—centre.

**Figure 10 materials-15-04912-f010:**
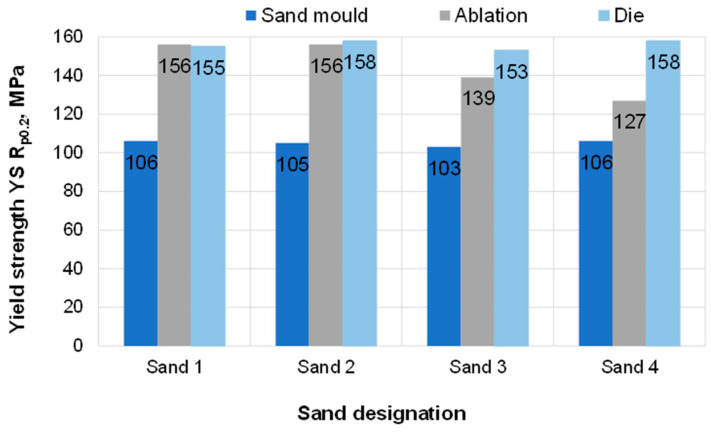
Yield strength Rp_0.2_ of the tested samples.

**Figure 11 materials-15-04912-f011:**
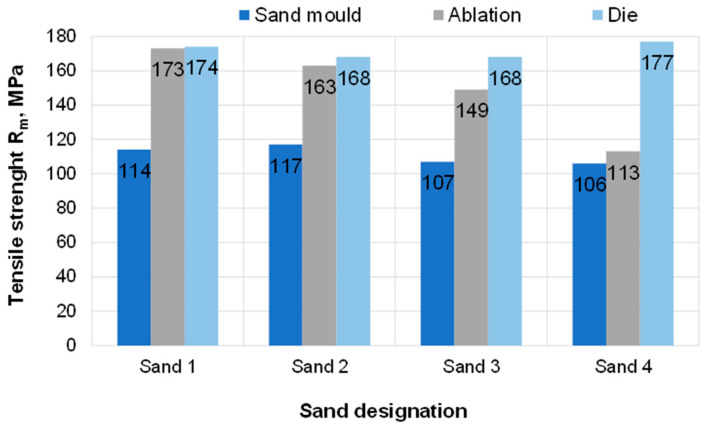
Tensile strength R_m_ of the tested samples.

**Figure 12 materials-15-04912-f012:**
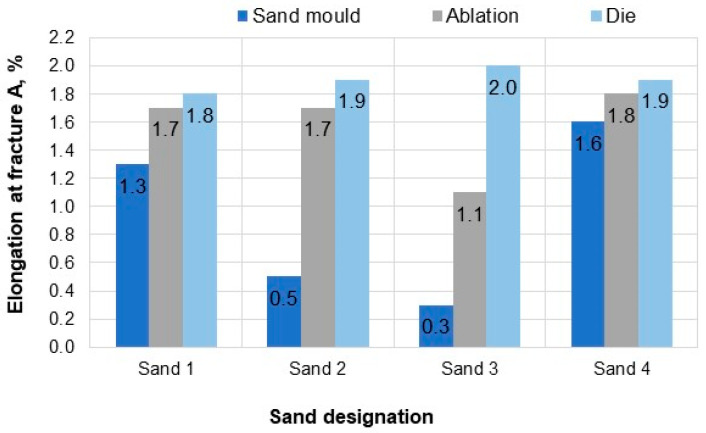
Elongation at fracture A of the tested samples.

**Table 1 materials-15-04912-t001:** Composition and parameters of the test sand mixtures.

Sand Designation	Sand Composition		R_g_^u^, MPa	Erosion Time, s
Moulding sand No. 1	silica sand phosphate binder (Glifos) hardener (MgO)	100 parts by mass 2.5 parts by mass 5% in relation to binder	1.64	11
Moulding sand No. 2	silica sand geopolymer binder (Geopol) thermally hardened	100 parts by mass 1.0 part by mass	1.68	11
Moulding sand No. 3	silica sand water glass R150 thermally hardened	100 parts by mass 1.0 part by mass	1.69	16
Moulding sand No. 4	silica sand geopolymer binder (Geopol) microwave-hardened	100 parts by mass 1.0 part by mass	1.70	23

**Table 2 materials-15-04912-t002:** SDAS, µm (50× magnification, oriented secants).

Sand Designation	Mould Type	Place in Casting			
		Top (G)	Centre (S)	Bottom (D)	Mean
**Sand No. 1**	Die	40.5	47.1	46.2	**44.6**
	Ablation	47.8	53.4	62.7	**54.6**
	Sand mould	68.6	74.6	60.9	**68.0**
**Sand No. 2**	Die	47.7	42.9	57.2	**49.3**
	Ablation	48.4	58.3	64.2	**56.9**
	Sand mould	72.2	62.7	82.2	**72.4**
**Sand No. 3**	Die	32.0	46.0	38.0	**38.7**
	Ablation	57.7	66.6	71.0	**65.1**
	Sand mould	80.0	86.0	80.0	**82.0**
**Sand No. 4**	Die	42.1	44.1	40.9	**42.4**
	Ablation	55.6	59.0	62.9	**59.2**
	Sand mould	68.8	75.7	74.9	**78.1**

**Table 3 materials-15-04912-t003:** Tensile test results obtained for specimens taken from the examined castings (average of 3 measurements).

Sand Designation	Mould Type	Rp_0.2_, MPa	R_m_, MPa	A, %	Z, %
**Sand No. 1**	Die	155	174	1.8	2.4
	Ablation	156	173	1.7	3.1
	Sand mould	106	114	1.3	1.3
**Sand No. 2**	Die	158	168	1.9	2.2
	Ablation	156	163	1.7	0.7
	Sand mould	105	117	0.5	0.9
**Sand No. 3**	Die	153	168	2.0	1.3
	Ablation	139	149	1.1	1.3
	Sand mould	103	106.7	0.3	-
**Sand No. 4**	Die	158	176.7	1.9	1.3
	Ablation	127	113.3	1.8	0.47
	Sand mould	106	106	1.6	-

## Data Availability

Data are contained within the article.

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
