# Peer review of "Effect of the Type of Inorganic Binder on the Microstructure and Properties of AlSi7Mg Alloy Castings Made by Ablation Casting Technology"

_materials, 2022, doi:10.3390/ma15144912_

Round 1
Reviewer 1 Report
1. The abstract is a little long, please consider to revise.
2. Line 71, “1.5 ÷ 1.7 MPa” should be “1.5-1.7MPa”. Line 173, “500 ÷ 800 µm” should be “500 - 800 µm”, please check the manuscript and revise it.
3. In line 97-104, the authors just present several results by others, however, the readers can hardly understand what is the real challenges of this technology.
4. Fig.2 should put in part 2, materials and methods.
5. The most important problem of this manuscript is that the authors just present some experimental results, which is lack of theoretical analysis. In other words, it looks like an experimental report rather than a scientific research. I suggest to reject for publication in present form, only if the authors rewrite it.
Author Response
Thank you very much for submitting comments on the article. the attachment contains the replies to the comments.

Reviewer 2 Report
The paper is very interesting for the aluminium casting industry and has a good potential. Overall, the paper should be corrected. in Introduction the lines 77 - 81 on page 2 would better fit to Materials and Methods section. The Materials and Method section is written in the way that I do not understand very well how the samples were cast. Four sand mixtures are described but I do not understand where the dies are taking place. You are presenting the results from the dies, but I do not understand were there four different dies used or only one for the comparison. Further, when you are presenting microstructures, you have then four microstructures from the dies...!? what is the difference in these dies? This part of experiment should be more clearly written.
In Table 1 the sand mixture nr. 2 and 3 seems the same, only the erosion time is different. Is the erosion time the parameter of the prepared sample or is it the result? If it is the result, it should be given in Results section.
Figures presenting microstructures (3 - 6) should be in same size and the gaps between pictures should be uniform for all figures. Also, the pictures are fogy and therefore clear.
One parameter not described in Experimental is the starting time of the ablation after the pouring. In figures the set up for temperature measurements is visible but you have no thermal analysis presented and belonging cooling rates. These are important for SDAS.
When describing microstructure constituents and Figures 7 and 8 you have mentioned microstructural constituents such as AlFeMgSi phase. You should use the correct stoichiometry or the name of the phase, in example with Greek alphabet – α-AlFeSi... In example what is c-Al(Fe, Mn).... Use subscripted numbers at Mg2Si...!
The second paragraph of Conclusions is not giving any conclusion only the hint of conclusion is there.
the 6th and 7th conclusion are not the case of this research, and they cannot be included in this paper!
The different solidification path is mentioned in conclusions but it is not exactly described in Results
Author Response

(The authors gave the same response as above.)

Reviewer 3 Report
Congratulations. The work is good, however, there are some concerns about your work, which can be resolved, improving the work and your understanding. Please see the attachment.

Author Response

(The authors gave the same response as above.)

Round 2
Reviewer 1 Report
The authors revised the paper according to the reviewer's comments. The reviewer believe it is suitable for publication.
Author Response
Thank you very much for re-reviewing the article and accepting the publication of the article in the Materials journal.
Reviewer 2 Report
Dear authors, the corrections seem to affect the quality of the work in a good way . I have doubts about the quality of the measurement of the temperature in Figs. 3a and 3c. The cooling curve of the ablation samples drops before you have started the ablation process. I think the measurement has failed (probably the thermocouple).
Also, the images showing the microstructures are still not aligned and it seems aucward.
Author Response
.
